# Effect modification of socioeconomic status in the association between contraception methods and couple's education: A secondary analysis of PDHS 2017–18

Sohail Lakhani[1]*, Syed Saqlain Ali Meerza[2], Shayan Khakwani[1], Syeda Kanza Naqvi[3], Maryam Hameed Khan[4], Muhammad Asim[1]

1 Department of Community Health Sciences, Aga Khan University, Karachi, Pakistan, 2 Postgraduate Medical Education, Aga Khan University, Karachi, Pakistan, 3 Institute for Global Health and Development, Aga Khan University, Karachi, Pakistan, 4 Bloomberg School of Public Health, Johns Hopkins University, Baltimore, Maryland, United States of America

* sohail.lakhani2@aku.edu, drsohailakhani@gmail.com

## Abstract

Globally, the region of South Asia reports the highest number of women (87 million) with unmet needs of contraception. Amongst the lower-middle-income countries of South Asia, Pakistan has performed poorly in enhancing contraceptive prevalence, as evident by the Contraceptive Prevalence Rate (CPR) of 34%. Factors including restricted access to contraception, a restricted selection of techniques, cultural/religious resistance, gender-based hurdles, and societal factors, such as the couple's education level, are among the most important causes for this gap in desire and usage. Thus, this study aimed to evaluate the association between couple's education level and their influence on their choice of contraception. In addition, the study also assessed the role of socioeconomic status in modifying the association between couple's education and contraception choice. Using PDHS 2017–18 data, couple's education status, preferences of contraceptive use and wealth quintiles were analyzed through multinomial logistic regression after adjusting for other confounding factors. The findings of our study revealed that out of the total sample of 14,368 women, 67.52% (n = 9701) were categorized as non-users, 23.55% (n = 3383) employed modern contraceptive methods, and 8.94% (n = 1284) utilized traditional contraceptive methods. Multivariable analysis showed that educated couples belonging to higher socioeconomic status (SES) had the highest adjusted odds ratio [7.66 (CI: 4.89–11.96)] of using modern contraceptives as opposed to uneducated couples of low socioeconomic statuses. Our analysis also revealed that the odds of using modern contraceptives were higher amongst mothers with five or more children [8.55 (CI:7.09–10.31)] as compared to mothers with less children when adjusted for other covariates. Thus, this study concludes the dynamic interplay between couple's level of education, contraceptive preference, and socioeconomic status This study contributes valuable insights for the policy makers and stakeholders to understand the intricate relationship between these factors.

**Data Availability Statement:** Data is available within the manuscript itself.

**Funding:** The authors received no specific funding for this work.

**Competing interests:** The authors have declared that no competing interests exist.

## Introduction

Over the last two decades, global demand for family planning has surged to almost 1.1 billion in 2020 from 900 million in 2000. Yet, about 270 million of the world's 1.9 billion reproductive-age women lack access to desired contraception. This gap is more pronounced in developing countries, where 234 million women seek pregnancy prevention without using any family planning method [1–3]. In higher-income nations, modern contraceptives met over 80% of the needs in 2019 (3). In contrast, lower-income regions, notably South Asia, struggle to meet half the demand, leaving 87 million women with unmet needs [3,4]. Factors include limited access, methods, cultural barriers, gender bias, demographics, and education gaps [5–10].

To promote health and well-being for all, every country should aim to amplify its efforts to promote the global availability of sexual and reproductive healthcare knowledge and supplies as outlined in 2030 Agenda for Sustainable Development Goals (3.7). This goal emphasizes integrating reproductive health into national strategies for comprehensive family planning and education [11]. However, progress in utilizing contemporary contraceptive techniques, as measured by SDG indicator 3.7.1, has stopped at 77% between 2015 and 2020 [1], emphasizing the need of identifying impediments to effective adoption.

Pakistan, a lower-middle-income nation of about 228 million people, ranks fifth globally in population, with a population surge of 57% since 1998 [12], driven by high fertility and stable mortality rates [13]. Despite being one of the first South Asian countries to launch a national family planning program, Pakistan has displayed poor performance in improving contraceptive prevalence [6]. As compared to other countries of the region, Pakistan reports the lowest contraceptive prevalence rate (CPR) accounting to 34% in year 2017 [14]. According to the most recent Annual Contraceptive Performance (ACP) Report 2019–20 by the Government of Pakistan, the overall contraceptive performance decreased by 8.3% compared to the previous year. The annual estimate of modern Contraceptive Prevalence Rate (mCPR) was identified as 41.0%, seeing a drop from 42.8% in the previous year [15]. As evident, the numbers have increased, however this is still well short of the CPR target of 50% by year 2020 in the Family Planning 2020 (FP2020) Commitment made by Pakistan at the 2012 London Summit [16].

Multiple factors drive these trends, including family dynamics, religious constraints, socioeconomic status, residence, and couple's education [17–19]. In Pakistan, studies reveal a positive link between contraceptive use and maternal age, parity, and living children [20,21]. This is rooted in limited autonomy for women in reproductive decisions, emphasizing prompt childbirth after marriage [5,20]. Literacy increases women's autonomy and contraceptive usage; particularly important is the wife's education, as educated women have fewer and well-educated children [22]. Households with low literacy and income are the most affected resulting in high fertility, and high childhood and maternal mortality rates [23,24]. Economic pressures drive a fertility rate of 3.8 births per woman, impacting maternal-child health [25]. Low literacy hampers awareness and use of contraception [26], while evidence links female literacy to lower infant mortality [27]. Household income also influences modern contraceptive use, with higher income group preferring modern contraceptive methods as opposed to traditional methods [28]. These factors highlight complex dynamics in contraceptive practices in Pakistan and so exploring the relationship between contraception use and education is vital to inform decisions resulting in healthy outcomes for mother and child [29].

To increase the uptake of contraceptive use in Pakistan, it is imperative to understand the various factors affecting the contraceptive usage preferences. The sizeable unmet need of contraception serves as the driving force to assess the dynamics between potential factors influencing the use of contraception in the socio-structural context of Pakistan. Thus, this study aimed to evaluate the association between couple's education level and their influence on their choice

of contraception. In addition, the study also assessed the role of socioeconomic status in modifying the association between couple's education and contraceptive choice.

## Methodology

### Sample data

The data for our secondary analysis was derived from the Pakistan Demographic and Health Survey (PDHS) of 2017–18, a large nationally representative survey of 15,068 women aged 15 to 49 years. The PDHS uses a multi-stage, clustered sampling strategy in which the number of clusters (enumeration areas as the primary sample units) from each region are chosen proportionally according to their size, and a predefined number of homes per cluster are chosen at random. PDHS collects data on reproductive history, family size preference, contraception methods used by the couple, employment status of the couple, residence (rural vs urban), education level of husband and wife, socioeconomic background characteristics, and health related decision capacity of married women as a standard practice.

### Study population

The sample our secondary analysis was restricted to ever married women only who had given complete information about all the covariates being considered for the study. Based on the above criteria, the final analysis was performed on a sample of 14,368 women.

### Variables of interest

**Outcome variable.** The outcome of interest was the preference of contraceptive use. It was a comparison of modern contraceptives with traditional methods and non-usage. Modern methods of contraception were defined as male condoms, female sterilization, male sterilization, injectables, pills, IUDs, implants, lactational amenorrhea method (LAM), Emergency contraception (EC), and standard days method (SDM). Traditional methods were defined as coitus interruptus withdrawal method, periodic abstinence and other (unspecified) local traditional practices.

**Independent variable.** The key primary exposure was the couple's education level, which was categorized into 4 subtypes; both husband and wife educated, only wife educated, only husband educated and both husband and wife uneducated. The variable was nominal in nature and was coded as '0' if neither the husband nor wife were educated, '1' if only the wife was educated, '2' if only the husband was educated, and '3' if both the husband and wife were educated. Education was defined as the husband/wife/both having education past primary level.

**Covariates.** Other variables used in the analysis included socioeconomic status in quintiles (poorest, poorer, middle income, richer, richest), employment status, women's age, husband's age, area of residence, health related decision making, number of children and beating justified. Justified beating was a binary composite variable which was created by using the responses of wives in which they were asked if their husbands were justified for beating when wives went out without telling their husband or when they were neglecting the children or when they were arguing with their husbands or if they burnt food or refused to have sex with husband. If women approved their beating justified in any of the conditions, they were coded as 1 'beating justified' and if they disapproved all the condition they were coded as 0 'not justified'.

**Effect modifier.** As part of our analysis, we also assessed if there was a relationship between contraceptive method usage and couple's education by their socioeconomic status.

The wealth index (socioeconomic status) originally computed by the DHS program using principal component analysis has five quintiles (poorest, poorer, middle, richer and richest).

## Statistical analysis

As part of descriptive analysis, frequencies and percentages were reported for all categorical variables including sociodemographic variables. Similarly, mean, and standard deviations were reported for all continuous variables including age of husband and wife. To investigate the relationship between contraceptive methods and independent factors, simple multinomial logistic regression was used to compute unadjusted odds ratio with 95% confidence intervals for both traditional and modern contraceptive usage when compared to non-users. To assess the multicollinearity, Cramer's V test was employed for two categorical variables and Eta test was employed for categorical and quantitative variables. Variables were entered into multivariable model based on their descending order of their F-statistics only if they were statistically significant at the univariable level. In addition, plausible effect modification of socioeconomic status was also assessed once the multivariable analysis revealed a parsimonious model. Adjusted odds ratio with 95% confidence intervals were reported for the final model. The level of significance was kept constant (p-value <0.05) at all levels of analysis, except for interaction analysis, where a p-value of 0.1 was considered significant. All the data was analyzed using STATA software version 17.

## Ethical statement

This study relies on secondary data of Pakistan Demographic Health Survey. This data is publicly and freely available on the DHS website (https://dhsprogram.com/data/available-datasets.cfm). Therefore, informed consent and ethical approval were not necessary. However, data access was formally obtained upon request from MEASURE DHS.

## Results

Among the total sample of 14,368 women, 67.52% (n = 9701) were categorized as non-users, 23.55% (n = 3383) were employing modern contraceptive methods, and 8.94% (n = 1284) were utilizing traditional contraceptive methods as shown in Fig 1. Among those utilizing modern methods, the most popular method was the application of male condoms (30.80%),

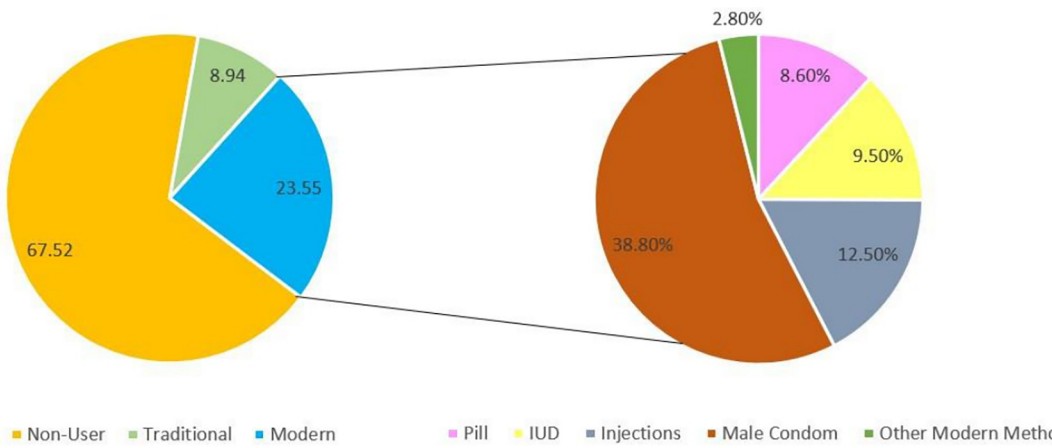

**Fig 1. Distribution of contraceptive usage by participants.**

followed by injections (12.50%), intrauterine devices (9.50%), pills (8.60%) and other modern methods (2.80%).

Table 1 displays the sociodemographic characteristics of women who have been married at least once, categorized according to their choice of contraceptive utilization. Among the total sample of 14,368 women, 67.52% (n = 9701) were categorized as non-users, 23.55% (n = 3383) were employing modern contraceptive methods, and 8.94% (n = 1284) were utilizing traditional contraceptive methods. The average age of women who were non-users of contraceptives was 30.78 years (SD = 8.50), while those employing traditional methods were found to have an average age of 34.29 years (SD = 7.29), and women utilizing modern methods had an average age of 34.28 years (SD = 7.27). In terms of husbands' ages, non-users' husbands had an average age of 35.84 years (SD = 9.92), husbands of traditional method users had an average age of 39.35 years (SD = 8.56), and husbands of modern method users had an average age of 39.60 years (SD = 8.72).

In cases where both spouses were educated, 28.56% chose modern methods, 11.39% chose traditional methods, and 60.05% did not use contraception. When both spouses were uneducated, 20.96% chose modern methods, 6.26% used traditional methods, and 72.78% used no contraception. Modern methods were more common in wealthier socioeconomic groups (29.64%) compared to lower-income groups (16.93%). In rural areas (n = 7,451), 22.77% used

**Table 1. Percentage distribution of ever-married women by choice of contraceptive used according to Pakistan DHS 2017–18.**

| Background characteristic | Outcome | | | |
|---|---|---|---|---|
| | Non-user (n = 9701) | Traditional (n = 1284) | Modern (n = 3383) | Number of women |
| **Couple's Education** | | | | |
| Both uneducated | 72.78 | 6.26 | 20.96 | 3,312 |
| Women educated | 64.06 | 10.19 | 25.75 | 608 |
| Men educated | 70.12 | 7.9 | 21.99 | 3,916 |
| Both educated | 60.05 | 11.39 | 28.56 | 6,532 |
| **Socioeconomic Status** | | | | |
| Poorest | 79.99 | 3.08 | 16.93 | 2,753 |
| Poorer | 71.11 | 6.39 | 22.50 | 3,062 |
| Middle | 63.28 | 9.88 | 26.83 | 2,825 |
| Richer | 61.52 | 10.87 | 27.61 | 2,746 |
| Richest | 55.51 | 14.85 | 29.64 | 2,982 |
| **Place of residence** | | | | |
| Urban | 57.48 | 13.8 | 28.72 | 6,917 |
| Rural | 70.65 | 6.58 | 22.77 | 7,451 |
| **Employment Status** | | | | |
| Not working and didn't work in last 12 months | 66.07 | 9.48 | 24.46 | 12,375 |
| Currently working | 64.51 | 8.19 | 27.29 | 1,993 |
| **Health related decision making** | | | | |
| Respondent alone | 65.96 | 9.23 | 24.81 | 1,363 |
| Not alone or others | 65.77 | 9.25 | 24.98 | 13,005 |
| **Beating Justified** | | | | |
| No | 62.26 | 10.91 | 26.82 | 7,422 |
| Yes | 70.36 | 7.08 | 22.56 | 6,946 |
| **Total Children Born** | | | | |
| 0–2 children | 83.37 | 5.35 | 11.28 | 5,955 |
| 3–4 children | 55.47 | 12.7 | 31.83 | 4,231 |
| 5 and more children | 50.54 | 11.42 | 38.04 | 4,182 |

modern contraceptives, while in urban areas (n = 6,917), this figure was 28.72%. Working women preferred modern methods more (27.29%) than non-working women (24.26%). Moreover, 24.81% of women with health-related decision-making authority and 26.82% of women who opposed husband's beating chose modern contraception.

Table 2 shows the univariate multinomial regression results. Factors associated with choice of contraceptive use at 5% level of significance included couple's education, place of residence, socioeconomic status, justified beating, total children ever born, and age of woman and of spouse. The odds of using modern contraceptives and traditional contraceptives as opposed to non-users amongst both educated spouses was 1.65 times and 2.21 times respectively as compared to both uneducated spouses. The odds of modern methods used compared to non-users were 55% more in urban as compared to rural residence. The likelihood of choosing modern methods was 2.52 times and traditional methods were 6.94 times the non-users amongst the wealthy socioeconomic group as compared to low-income group. Women who thought beating was never justified preferred modern methods 34% more as compared to those who thought beating was justified. As the number of children increased, the odds increased from 4.24 to 5.56 times when compared to 0–2 children for mothers having 3–4 and 5 and above

**Table 2. Results of univariate multinomial logistic regression analysis.**

| Variables | Traditional method | Modern method |
|---|---|---|
| | Unadjusted OR (CI) | Unadjusted OR (CI) |
| **Couple's education** | | |
| Both uneducated | Ref | Ref |
| Only women educated | 1.85 (1.21–2.82) | 1.39(1.05–1.85) |
| Only men educated | 1.31 (1.04–1.64) | 1.08 (0.92–1.27) |
| Both educated | 2.21 (1.67–2.92) | 1.65 (1.38–1.97) |
| **Place of residence** | | |
| Rural | Ref | Ref |
| Urban | 2.58 (1.99–3.32) | 1.55 (1.33–1.80) |
| **Socioeconomic status** | | |
| Poorest | Ref | Ref |
| Poorer | 2.33 (1.60–3.39) | 1.49 (1.19–1.87) |
| Middle | 4.05 (2.69–6.08) | 2.00 (1.57–2.54) |
| Richer | 4.58 (3.12–6.73) | 2.12 (1.67–2.68) |
| Richest | 6.94 (4.77–10.09) | 2.52 (1.94–3.26) |
| **Employment status** | | |
| Currently working | Ref | Ref |
| Not working and didn't work in the last 12 months | 1.13 (0.86–1.47) | 0.88 (0.74–1.02) |
| **Health related decision making** | | |
| Not alone or others | Ref. | Ref. |
| Respondent alone | 0.99 (0.73–1.36) | 0.99 (0.80–1.22) |
| **Beating Justified** | | |
| Justified | Ref. | Ref. |
| Not Justified | 1.74 (1.39–2.17) | 1.34 (1.17–1.53) |
| **Total children ever born** | | |
| 0–2 children | Ref. | Ref. |
| 3–4 children | 3.56 (2.97–4.27) | 4.24 (3.58–5.02) |
| 5 and above | 3.52 (2.75–4.50) | 5.56 (4.60–6.72) |
| **Woman's age** | 1.05 (1.04–1.06) | 1.05 (1.04–1.06) |
| **Spouse's age** | 1.04 (1.03–1.05) | 1.04 (1.03–1.05) |

children respectively. With every 1-year increase in the age of wife and husband, the odds of using modern methods increased by 1.05 and 1.04 times respectively.

Table 3 displays adjusted odds ratios along with 95% confidence intervals of variables significantly associated with the choice of contraceptives used. The odds of using traditional contraceptive methods versus no contraceptive users amongst both husband and wife educated belonging to a poorer socioeconomic status is 3.27 times the odds amongst both spouses uneducated belonging to a poorest socioeconomic status when adjusted for all other variables. Similarly, the odds of using traditional contraceptive methods versus no contraceptive users amongst both husband and wife educated belonging to a middle socioeconomic status is 4.23 times the odds amongst both spouses uneducated belonging to a poorest socioeconomic status when adjusted for all other variables. These varying odds at different levels of exposure explain the effect-modification (interaction effect) of socioeconomic status on the preferred contraceptive method.

**Table 3. Results of multivariable multinomial logistic regression.**

| Variables | Traditional method | Modern method |
|---|---|---|
| | Adjusted OR (CI) | Adjusted OR (CI) |
| **Couple education** | | |
| Both uneducated | Ref. | Ref. |
| Only women educated | 0.32 (0.04–2.50) | 1.09 (0.41–2.92) |
| Only men educated | 0.49 (0.25–0.96) | 0.89 (0.66–1.22) |
| Both educated | 4.02 (1.61–10.06) | 2.23 (1.37–3.64) |
| **Total children ever born** | | |
| 0–2 children | Ref. | Ref. |
| 3–4 children | 3.96 (3.26–4.80) | 4.70 (3.95–5.59) |
| 5 and above | 5.82 (4.48–7.56) | 8.55 (7.09–10.31) |
| **Place of residence** | | |
| Rural | Ref. | Ref. |
| Urban | 1.63 (1.19–2.20) | 1.17 (0.97–1.41) |
| **Socioeconomic status** | | |
| Lowest | Ref. | Ref. |
| Lower | 2.27 (1.35–3.81) | 1.49 (1.09–2.23) |
| Middle | 4.59 (2.54–8.31) | 2.28 (1.52–3.42) |
| Higher | 2.42 (1.11–5.22) | 1.79 (1.05–3.03) |
| Highest | 3.41 (1.16–10.0) | 1.72 (0.77–3.86) |
| **Couple's education # SES** | | |
| Both uneducated and Poorest | Ref. | Ref. |
| Only women educated and Poorer | 2.40 (0.95–6.02) | 3.06 (2.12–4.39) |
| Only women educated and Middle | 5.91 (2.94–11.80) | 2.46 (1.51–3.99) |
| Only women educated and Richer | 6.08 (3.02–12.26) | 4.05 (2.50–6.53) |
| Only women educated # Richest | 8.34 (3.29–21.13) | 2.54 (1.43–4.50) |
| Only men educated # Poorer | 2.26 (1.42–3.58) | 1.74 (1.12–2.67) |
| Only men educated # Middle | 3.38 (2.16–5.27) | 2.06 (1.36–3.10) |
| Only men educated # Richer | 4.66 (2.80–7.73) | 2.08 (1.34–3.21) |
| Only men educated # Richest | 5.67 (3.48–9.23) | 1.46 (0.93–2.30) |
| Both educated # Poorer | 3.27 (1.89–5.62) | 2.19 (1.52–3.13) |
| Both educated # Middle | 4.23 (2.72–6.55) | 3.17 (1.63–6.13) |
| Both educated # Richer | 5.41 (3.54–8.25) | 3.87 (1.96–7.60) |
| Both educated # Richest | 7.66 (4.89–11.96) | 4.60 (1.97–10.73) |

It is important to understand that the odds of modern contraceptive use as compared to no use is 8.55 times amongst women with 5 or more children as opposed to women with zero to two kids when adjusted for all other variables. The odds of traditional contraceptive use as compared to no use is 5.82 times amongst women with 5 or more children as opposed to women with zero to two kids when adjusted for all other variables.

## Discussion

The findings of our study depicted that educated couples were more likely to use contraceptives, both traditional and modern contraceptives. In addition, socioeconomic status modified the relationship between contraceptive preference and couple's education after accounting for other factors. These findings support the hypothesis that education positively influences contraceptive use.

In line with the majority of South Asian and international studies, our research demonstrated that women's educational status exerted an influence on contraceptive usage, indicating that higher levels of education correlated with increased utilization of contraceptives, particularly favoring modern techniques [30–32]. Additionally, it has been noted that literacy amplifies women's autonomy and their ability to make decisions, consequently promoting higher adoption rates of contraceptives. Moreover, this heightened autonomy through literacy has a positive impact on women's reproductive choices and family planning decisions [23]. These findings underscore the crucial role of women's education, enabling them to postpone marriage and childbirth while making well-informed decisions about the use of modern contraceptives, setting them apart from their less-educated counterparts [33].

We also observed that husband's education in addition to women's education increases the odds of modern contraceptive use compared to those who are uneducated. Our findings corroborated with studies done in Ethiopia, which showed that a woman married to an uneducated man was 72% less likely to use modern contraceptive methods compared to a woman married to a man who had completed at least the 7th grade [34]. The increase in modern contraceptive utilization among educated husbands can be attributed to the dynamics of patriarchal societies, such as Pakistan, where men often hold the primary decision-making power regarding reproduction. In such contexts', educated husbands were more inclined to engage in discussions about modern family planning methods [35].

The study also concluded that SES was a significant predictor of contraceptive use and modified the role of education on contraceptives preference. The findings of our study were validated by previous studies which had observed the likelihood of contemporary contraception usage being greater among wealthier women than among the impoverished [36–38]. This interaction between education and SES may be explained by the fact that as education levels rise, so do wealth and reputation, as well as the goal to restrict children via the use of contraception [39]. Evidently, education increases one's potential to obtain money and reputation; this clashes with the choice to have children since, in a contemporary economy, children are capital consumers instead of creators for many generations [39].

On the other hand, our results showed that highly educated, non-poor couples preferred the use of traditional contraceptive methods over modern methods. The topic of family planning and sexuality has remained a taboo in Asian countries, leading to embarrassment among people who consider using modern contraceptive methods. This embarrassment stems from various factors, including religious beliefs, fear of side effects, conformity to society norms, and lack of awareness [40]. Some educated couples adhere to cultural and religious beliefs that favor the use of traditional contraceptives as they may perceive modern contraceptives, as conflicting with their religious principles [41]. Research findings indicate that a significant

number of couples feel hesitant to discuss their preference for modern contraceptives with their partners, particularly women who may avoid using oral contraceptive pills due to concerns about potential social judgment labeling them as promiscuous [40–42]. Finally, a lack of comprehensive family planning education, even within educated couples, has contributed to the increasing prevalence of traditional contraceptive methods.

The study's findings also highlight the significant influence of the number of children ever born on the likelihood of contraceptive use as opposed to non-use. We found similar findings present in China and Bangladesh where the number of living children increases contraceptive use [43,44]. This suggests that women, upon achieving their desired number of children, employ contraception with the intention of avoiding pregnancy rather than for the purpose of birth spacing or diminishing their desired family size. Similarly, the likelihood of using contraception increased among those couples living in urban areas than those of their rural counterparts. However, we found an increase of traditional contraceptive methods among urban residents contrary to previous studies [45,46]. One plausible explanation for this contradiction is the fear of side effects among modern contraceptive methods, such as birth control pills. The recurring concern of 'side effects' is a prevalent issue in developing nations like Pakistan, India, and various African countries, acting as a frequently encountered obstacle to the adoption of modern contraception [47–49].

There are several strengths in this study. The utilization of the nationwide DHS survey offered a relatively large sample size, producing accurate estimations along with their standard errors and enabling us to generalize the results for countrywide policy making and interventions. The study's innovation is also evident in its approach of monitoring contraceptive usage patterns by incorporating the couple's education level as a primary variable, a unique aspect not explored in any previous research related to Pakistan. However, significant shortcomings of the research were also found. The design of the research (cross-sectional) does not permit us to demonstrate causation among parameters. Furthermore, since we utilized secondary data, additional drivers such as level of services, including counseling about adverse reactions and supply availability, and contextual elements were not incorporated into the design. To address the shortcomings related to the cross-sectional design and the lack of several crucial variables in this investigation, it is advised to conduct a longitudinal study using primary data to collect evidence on all crucial aspects linked with the use of contemporary contraceptives.

This study also has several limitations. Firstly, the dataset used is cross-sectional, which raises concerns regarding temporality and causality. These are inherent limitations of such designs, as they cannot establish causal relationships. Secondly, although we controlled for several potentially confounding variables, there remains a possibility of confounding due to other unidentified factors. Thirdly, residual confounding stemming from the variables included in our multivariable multinomial logistic regression analysis may also persist.

## Conclusion

It was observed that couple's education, area of residence, and number of children alive were most important factors in predicting the use of contraceptives. Socio-economic status was an effect modifier of education of couple and their choice of contraceptive method. The findings of the study will be useful to inform future programs and policy interventions that would assist couples to meet their reproductive goals and contribute to the country's self-sufficiency in different human development indicators.

## Acknowledgments

We have no acknowledgments to declare.

## Author Contributions

**Conceptualization:** Sohail Lakhani, Syed Saqlain Ali Meerza, Syeda Kanza Naqvi.

**Data curation:** Sohail Lakhani, Syeda Kanza Naqvi.

**Formal analysis:** Sohail Lakhani.

**Investigation:** Sohail Lakhani, Maryam Hameed Khan.

**Methodology:** Sohail Lakhani, Maryam Hameed Khan.

**Project administration:** Sohail Lakhani, Muhammad Asim.

**Resources:** Sohail Lakhani.

**Software:** Sohail Lakhani, Shayan Khakwani.

**Supervision:** Sohail Lakhani, Muhammad Asim.

**Validation:** Sohail Lakhani, Maryam Hameed Khan, Muhammad Asim.

**Visualization:** Sohail Lakhani, Shayan Khakwani, Muhammad Asim.

**Writing – original draft:** Sohail Lakhani, Syed Saqlain Ali Meerza, Shayan Khakwani, Maryam Hameed Khan.

**Writing – review & editing:** Sohail Lakhani, Syed Saqlain Ali Meerza, Shayan Khakwani, Muhammad Asim.

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
