## [Decision Letter · Decision Letter 0]

23 Feb 2024

PGPH-D-23-01854

Effect modification of socioeconomic status in the association between contraception methods and couple’s education: a secondary analysis of PDHS 2017-18

Dear Dr. Lakhani,

Thank you for submitting your manuscript to PLOS Global Public Health. After careful consideration, we feel that it has merit but does not fully meet PLOS Global Public Health’s publication criteria as it currently stands. Therefore, we invite you to submit a revised version of the manuscript that addresses the points raised during the review process.

The reviewers both suggested minor revisions alongside their acceptance recommendations for this manuscript, which we feel would be beneficial to improve the study reporting and final publication. Please highlight the strengths and limitations of the study within your discussion, ensure references are up to date and find relevant recent references where possible, and consider discussing why educated couples used traditional methods of contraception.

We look forward to receiving your revised manuscript.

Kind regards,

Jennifer Tucker, PhD

Associate Editor

Journal Requirements:

2. Tables should not be uploaded as individual files. Please remove these files and include the Tables in your manuscript file as editable, cell-based objects. For more information about how to format tables, see our guidelines: 

https://journals.plos.org/globalpublichealth/s/tables

3. In the online submission form, you indicated that "Data will be made available upon request". All PLOS journals now require all data underlying the findings described in their manuscript to be freely available to other researchers, either 1. In a public repository, 2. Within the manuscript itself, or 3. Uploaded as supplementary information.

Additional Editor Comments (if provided):

Reviewers' comments:

Reviewer's Responses to Questions

**Comments to the Author**

1. Does this manuscript meet PLOS Global Public Health’s publication criteria? Is the manuscript technically sound, and do the data support the conclusions? The manuscript must describe methodologically and ethically rigorous research with conclusions that are appropriately drawn based on the data presented.

Reviewer #1: Yes

Reviewer #2: Yes

2. Has the statistical analysis been performed appropriately and rigorously?

Reviewer #1: Yes

Reviewer #2: Yes

3. Have the authors made all data underlying the findings in their manuscript fully available (please refer to the Data Availability Statement at the start of the manuscript PDF file)?

Reviewer #1: Yes

Reviewer #2: Yes

4. Is the manuscript presented in an intelligible fashion and written in standard English?

Reviewer #1: Yes

Reviewer #2: Yes

5. Review Comments to the Author

Reviewer #1: This study evaluates the association between a couple’s education level and their influence on their choice of contraception. In addition, the study also assessed the role of socioeconomic status in modifying the association between a couple’s education and contraception choice. Using PDHS 2017-18 data, the couple’s education status, preferences of contraceptive use, and wealth quintiles were analyzed through multinomial logistic regression after adjusting for other confounding factors.

findings from this study revealed that out of the total sample of 14,368 women, 67.52% (n = 9701) were categorized as non-users, 23.55% (n = 3383) employed modern contraceptive methods, and 8.94% (n = 1284) used traditional contraceptive methods. Multivariable analysis showed that educated couples belonging to higher socioeconomic status (SES) had the highest adjusted odds ratio [7.66 (CI:4.89-11.96)] of using modern contraceptives as opposed to uneducated couples of low socioeconomic statuses. The odds of using modern contraceptives were higher amongst mothers with five or more children [8.55 (CI:7.09-10.31)] as compared to mothers with fewer children when adjusted for other covariates.

Thus, this study concludes the dynamic interplay between a couple’s level of education, contraceptive preference, and socioeconomic status This study contributes valuable insights for the policymakers and stakeholders to understand the intricate relationship between these factors.

Overall, this is a good study. It contributes to the knowledge of the uptake of contraception in developing countries. It is well-written and technically sound. I have just a few concerns written below that need to be addressed.

1) The strengths and limitations of the study should be highlighted.

2) Most of the references are too old.

Reviewer #2: it is an excelent and innovative analysis that could inspire others to use it. Congratulations to the authors.

may be would be good the explain why educated couples used traditional methods of contraception, or at least try to understand why.

6. PLOS authors have the option to publish the peer review history of their article (what does this mean?). If published, this will include your full peer review and any attached files.

**Do you want your identity to be public for this peer review?** For information about this choice, including consent withdrawal, please see our Privacy Policy.

Reviewer #1: No

Reviewer #2: No

---

## [Editor Report · Decision Letter 1]

10 Jun 2024

Effect modification of socioeconomic status in the association between contraception methods and couple’s education: a secondary analysis of PDHS 2017-18

PGPH-D-23-01854R1

Dear Dr Lakhani,

We are pleased to inform you that your manuscript 'Effect modification of socioeconomic status in the association between contraception methods and couple’s education: a secondary analysis of PDHS 2017-18' has been provisionally accepted for publication in PLOS Global Public Health.

Best regards,

Olaide Olawumi Ojoniyi, PhD

Guest Editor